# Resting-State Functional Connectivity Difference in Alzheimer’s Disease and Mild Cognitive Impairment Using Threshold-Free Cluster Enhancement

**DOI:** 10.3390/diagnostics13193074

**Published:** 2023-09-28

**Authors:** Ramesh Kumar Lama, Goo-Rak Kwon

**Affiliations:** Department of Information and Communication Engineering, Chosun University, 309 Pilmundaero, Gwangju 61452, Republic of Korea; rklama@chosun.kr

**Keywords:** Alzheimer’s disease, default mode network, large-scale brain network, functional connectivity, threshold-free cluster enhancement

## Abstract

The disruption of functional connectivity is one of the early events that occurs in the brains of Alzheimer’s disease (AD) patients. This paper reports a study on the clustering structure of functional connectivity in eight important brain networks in healthy, AD, and prodromal stage subjects. We used the threshold-free cluster enhancement (TFCE) method to explore the connectivity from resting-state functional MR images (rs-fMRIs). We conducted the study on a total of 32 AD, 32 HC, and 31 MCI subjects. We modeled the brain as a graph-based network to study these impairments, and pairwise Pearson’s correlation-based functional connectivity was used to construct the brain network. The study found that connections in the sensory motor network (SMN), dorsal attention network (DAN), salience network (SAN), default mode network (DMN), and cerebral network were severely affected in AD and MCI. The disruption in these networks may serve as potential biomarkers for distinguishing AD and MCI from HC. The study suggests that alterations in functional connectivity in these networks may contribute to cognitive deficits observed in AD and MCI. Additionally, a negative correlation was observed between the global clinical dementia rating (CDR) score and the Z-score of functional connectivity within identified clusters in AD subjects. These findings provide compelling evidence suggesting that the neurodegenerative disruption of functional magnetic resonance imaging (fMRI) connectivity is extensively distributed across multiple networks in individuals diagnosed with AD.

## 1. Introduction

Alzheimer’s disease (AD) is the primary cause of dementia, commonly affecting people over 65 years old [1,2]. The accumulation of amyloid-β and tau neurofibrillary tangles in the brain is a crucial factor in the death of neurons and the breakdown of neural pathways [3,4,5]. This degenerative process is marked by short-term memory loss, cognitive dysfunction, language impairments, and difficulty with executive functions. Studies of degeneration phenomena in the brain are often conducted using large-scale network models [6]. These studies have shown that decreases in brain volume and alterations in connectivity are prevalent in both AD and mild cognitive impairment (MCI) subjects. A range of biomarkers are employed to evaluate the progression of Alzheimer’s disease. These biomarkers are measurable indicators of various biological processes and are utilized to diagnose and monitor this neurodegenerative disorder [6]. Different imaging models, such as magnetoencephalography, electroencephalography, and functional magnetic resonance imaging (fMRI), are used to study the connectivity patterns in AD and MCI brains [7]. These studies have found that the disruption of functional connectivity is a significant feature of AD and MCI. Additionally, studies have shown that networks that are active during the passive or resting state of the brain are disrupted. These networks include the default mode network (DMN), central executive network (CEN), and salience network (SN) [8,9,10]. Although changes are often seen in the DMN, SN, and CEN across the spectrum of AD and MCI, resting-state functional magnetic resonance imaging (rs-fMRI) results have shown that individuals with aging or MCI also exhibit functional connectivity alterations in these large-scale networks. Recent studies have demonstrated that these are not only visible in the DMN but also in salience, motor, sensory motor, dorsal attention, auditory, and visual networks. Agosta used resting-state functional magnetic resonance imaging (rs-fMRI) to explore connectivity patterns in the brains of Alzheimer’s disease (AD) patients, amnestic mild cognitive impairment (aMCI) patients, and healthy controls [11,12]. The study found that AD patients had decreased connectivity in the default mode network (DMN) and enhanced connectivity in frontal networks compared to controls and aMCI patients. The only abnormality found in aMCI patients was a reduction in precuneus connectivity in the DMN. The changes in connectivity were only partly related to gray matter atrophy. In AD patients, increased connectivity in the executive network was associated with better frontal-executive and language neuropsychological scores. These results suggest that AD is associated with alterations in large-scale functional brain networks beyond the DMN, and changes in connectivity may be an attempt to maintain cognitive efficiency. Additionally, a change in the medial parietal rs-fMRI signal seems to be present in the early phase of AD. Aoki et al. [13] conducted a study using EEG resting-state networks to examine changes in the DMN in individuals with AD and mild cognitive impairment due to AD (ADMCI). The study employed eLORETA-ICA [13], a method for evaluating the activities of five EEG resting-state networks, in a cohort of 90 drug-free AD patients, 11 drug-free subjects with MCI, and 147 healthy control subjects. The results revealed a significant decrease in activities within the memory network and occipital alpha activity in the AD/ADMCI group. Linear regression analysis was used to account for confounding factors, and the age-corrected EEG-RSN activities showed correlations with cognitive function test scores in the AD/ADMCI group. Specifically, reduced activity in the memory network was strongly associated with lower cognitive scores on the MMSE and ADAS-J cog tests, particularly in orientation, registration, repetition, word recognition, and ideational praxis. These findings suggest that specific EEG resting-state networks are selectively affected by AD, and the decline in network activity contributes to the manifestation of AD symptoms. The study also highlights the usefulness of eLORETA-ICA as a non-invasive tool for assessing EEG-functional network activities, providing insights into the complex neurophysiological mechanisms underlying AD.

Additionally, fMRI has been used to investigate the functional connectivity of different subregions of the amygdala AD patients compared to healthy controls [14]. Three subregions of the amygdala were defined based on probabilistic cytoarchitectonic atlases, and their whole-brain resting-state functional connectivity was mapped. The study found disrupted connectivity patterns in the lateral basal nucleus of the amygdala in AD patients compared to controls, which predicted disconnection with various brain regions. These findings suggest that different subregions of the amygdala may have distinct connectivity patterns and contribute differently to cognitive deficits in AD. Furthermore, a study using fMRI investigated the functional connectivity of the DMN in patients with AD compared to healthy controls using independent component analysis (ICA) and Bayesian network (BN) techniques [15]. The study found decreased resting-state functional connectivity in the DMN of AD patients compared to HC, which was consistent with previous studies. The study also revealed altered effective connectivity in AD, including lost connections from the left hippocampus to the left inferior parietal cortex, left inferior temporal cortex, medial prefrontal cortex, and posterior cingulate cortex, as well as changes in connection directions between certain regions. These findings suggest that altered effective connectivity in AD may serve as a potential biomarker and reveal more characteristics of the disease. A recent study found that exercise training can improve within- and between-network connectivity in older individuals with intact cognition and MCI due to Alzheimer’s disease [16]. Another study used a computational brain network model to link neuronal hyperactivity to large-scale oscillatory slowing in early-stage Alzheimer’s disease [17]. A machine learning study found altered large-scale dynamic connectivity patterns in AD and MCI patients [18]. Additionally, a triple-network dynamic connection study demonstrated that AD is associated with the abnormal organization and functioning of large-scale brain networks [19]. Therefore, we included other networks in our investigation of brain connectivity disruptions. Most studies use threshold-based statistical methods to generate edgewise significance values, defining an a priori clustering threshold. Network-based statistics (NBS) [20,21] operate by thresholding the network connections and identifying connected components or subnetworks that exhibit significant differences in connectivity. In contrast, the threshold-free cluster enhancement method (TFCE) [22,23] generates edgewise significance values without the need for an a priori definition of a hard edge-defining threshold. Consequently, TFCE offers a more insightful view into the altered cluster structure of brain connectivity. In this study, we employed TFCE to identify the significant differences in connectivity patterns between groups.

We hypothesize that individuals with AD and MCI exhibit disrupted functional connectivity within key brain networks, including the SMN, DAN, SAN, and DMN. This disruption, characterized by reduced interregional correlations and connectivity patterns, may serve as a potential biomarker for distinguishing AD and MCI from healthy controls. The hypothesis further posits that the observed impairments within the DMN, a network associated with episodic memory processing and cognitive rest, contribute significantly to the cognitive deficits observed in AD and MCI individuals. To test this hypothesis, we divided the whole brain into 164 different anatomical regions of interest (ROIs) and used eight important large-scale networks comprising only 32 ROIs. Figure 1 shows the schematic of the data analysis pipeline of the proposed study. TFCE is used to identify the abnormal subnetworks between two subject groups: AD versus HC and HC versus MCI. ANOVA is performed to identified clusters between all three groups AD, HC, and MCI.

## 2. Materials and Methods

This article was prepared using data obtained from the Alzheimer’s Disease Neuroimaging Initiative database (ADNI) (http://adni.loni.usc.edu/) [24]. Launched in 2003 under the leadership of principal investigator Michael W. Weiner, MD, ADNI is a public–private partnership. Its main objective has been to determine if the combination of serial magnetic resonance imaging (MRI), positron emission tomography (PET), other biological markers, and clinical and neuropsychological assessments can effectively measure the progression of MCI and early AD. The imaging scanning protocol utilized in this study remains consistent with the protocol described in our previously published article [25]. The subjects for this study were selected according to the criteria outlined in Table 1.

### 2.1. Subjects

Ninety-five subjects were selected from the ADNI database. We selected the subjects according to the availability of both structural MRI (sMRI) and fMRI imaging data. sMRI is a T1-weighted sequence that provides high-resolution anatomical images of the brain. Subjects with the following demographic status were selected in our study out of all the available data in the ADNI database. The HC group consists of 31 subjects with 14 males and 17 females, the MCI group consists of 17 females and 14 males, and the AD group consists of 15 males and 18 females.

### 2.2. Data Preprocessing

We processed the fMRI and sMRI images using the CONN toolbox [26], which uses a default pipeline for preprocessing. This pipeline includes the realignment and unwrapping of functional data, slice timing correction, outlier identification, direct segmentation and normalization, and functional smoothing. To realign the functional data, the CONN toolbox uses SPM12 realign [27] and unwrap procedures [26], with b-spline interpolation used to co-register and resample all scans to a reference image. To correct temporal misalignment between different slices of functional data, the slice timing correction (STC) procedure of SPM is used [28]. In addition, to identify outlier scans, the Artefact Detection Tools (ART) toolbox (https://www.nitrc.org/projects/artifact_detect/) is used in CONN. The ART toolbox identifies outlier scans based on the observed global bold signals and amount of subject motion in the scanner. Outlier scans were identified as global bold signals exceeding five standard deviations from the global mean or with frame-wise displacement above 0.9 mm.

The next step involved normalizing the functional and anatomical data into the standard MNI space. This was carried out by segmenting the functional and anatomical data into gray matter, white matter, and cerebrospinal fluid (CSF) classes using the SPM12 unified segmentation method [29]. Outlier detection was performed after normalization and segmentation. The difference image of the mean BOLD signal was used as a reference image for the functional data, while the T1-weighted volume was used as a reference image for the structural data [29,30]. In order to enhance the signal-to-noise ratio and reduce residual variability in functional and gray anatomy data across subjects, a resampling technique was employed. Both the functional and anatomical data were resampled to a bounding box measuring 180 × 216 × 180 mm, with voxel sizes of 2 mm isotropic for functional data and 1 mm for anatomical data. This resampling was accomplished using fourth-order spline interpolation. Subsequently, the functional data underwent smoothing via spatial convolution with an 8 mm full width at half maximum (FWHM) Gaussian kernel.2.3.

### 2.3. Brain Connectivity Estimation

In the CONN toolbox, a seed-based resting-state functional connectivity (FC) analysis was undertaken using a total of 164 regions of interest (ROIs). The initial step, known as the first-level analysis, assessed individual subjects by examining the connectivity between each of the 164 seeds and other brain voxels, utilizing Pearson’s correlation coefficient for this purpose. Following this, a group-level or second-level analysis contrasted conditions. After the individual analyses, ROI analyses employed either F-statistics or Wilks lambda statistics. To ensure the correlation values were suitable for parametric statistical testing at the group level, Fisher’s transformation was applied. Of the 164 ROIs used, 132 were sourced from the FSL Harvard-Oxford atlas, which includes both cortical and subcortical regions, and cerebellar areas derived from the AAL atlas. The remaining 32 ROIs represented various networks such as the DMN, sensorimotor, visual, salience, dorsal attention, frontoparietal, language, and cerebellar networks. In the analysis, these ROIs, covering both the resting-state networks and the atlas regions, were combined to differentiate functional connectivity pairs.

The functional connectivity among different ROIs was estimated on the basis of temporal correlations of BOLD signals in these regions. Initially, we extracted the BOLD time series from 164 regions. Among these regions, only 32 ROIs were selected to construct 8 important large-scale networks: default mode (4 ROIs), sensorimotor (3 ROIs), visual (4 ROIs), salience/cingulo-opercular (7 ROIs), dorsal attention (4 ROIs), frontoparietal/central executive (4 ROIs), language (4 ROIs), and cerebellar (2 ROIs). Consistent with previous studies, Pearson’s correlation was estimated among the pairs of regions and expressed in terms of an ROI-to-ROI correlation matrix.
(1)$$r(i,j)=\frac{\int R_{{}_{i}}(t)R_{j}(t)}{(\int R_{i}^{2}(t)dt\int R_{j}^{2}(t)dt{)}^{1/2}}$$
where *R* is the BOLD time series within each ROI (for simplicity, all-time series are considered to be centered at zero mean), *r* is a matrix of correlation coefficients. We estimated the RRC symmetric matrix of Fisher-transformed correlation coefficients as
*Z*(*i*, *j*) = tanh^−1^(*r*(*i*, *j*))(2)

## 3. Statistical Analysis

We used threshold-free cluster enhancement (TFCE) of the CONN toolbox to identify the differences in the brain networks between AD and HC and between MCI and HC.

### Threshold-Free Cluster Enhancement (TFCE)

NBS is a parameter-free statistical tool used to identify group differences when the distribution is known. The first step of NBS is to estimate the connectivity matrix *r*, which is of size *N* × *N*. Each element of this matrix, *r*(*i*, *j*), represents the edge value between two regions of interest (ROIs) *i* and *j*. After estimating the connectivity matrix, the second step of NBS is to apply a threshold on the edge values *r*(*i, j*) to define a cluster. Next, the thresholded connections are grouped for each subject belonging to the defined group. Finally, a permutation test is performed on these edge values, repeating the process for *n* different combinations. We count the number of test statistics whose value is greater than the initial test statistics, and the *p*-value is calculated by estimating the ratio of this number to the total number of test statistics. In contrast, TFCE performs an enhancement operation for each edge value *r*(*i*, *j*) instead of a thresholding operation. We start with an ROI-to-ROI connectivity matrix, which is derived from a General Linear Model analysis. ROIs are sorted automatically using a hierarchical clustering procedure, which considers anatomical proximity or functional similarity metrics. Instead of using a fixed height threshold as in NBS, TFCE computes a TFCE score map [20,30]. The TFCE score map combines the strength of the statistical effect for each connection with the extent of neighboring connections that show similar effects.
(3)$$TFCE(i,j)=\int \begin{array}{l} h(i,j)\\ h=h_{0} \end{array}e{\left(h\right)}^{E}h^{H}dh$$
where *e*(*h*) represents the extension of neighboring connections and *h* represents the height of the fractional edge value. *E* and *H* are extension and height enhancement parameters, respectively. This helps capture both local and distributed effects. The expected distribution of TFCE values under the null hypothesis is estimated using permutation iterations of the original data using 1000 iterations. For each cluster in the original analysis, a peak-level Family-Wise Error (FWE)-corrected *p*-value is computed. This indicates the likelihood under the null hypothesis of observing at least one or more connections with the given TFCE scores over the entire ROI-to-ROI connectivity matrix. For the peak-level analysis, local peaks in the TFCE score map are compared to the null hypothesis distribution of local-peak TFCE values. This estimation provides *p*-values for each peak, representing the likelihood under the null hypothesis of observing a peak with similar or larger scores by chance. In small sample sizes, the correction for multiple comparisons becomes increasingly important as the number of comparisons rises. However, with such sizes, the application of correction methods might be overly conservative, increasing the risk of false negatives. Therefore, we chose to use uncorrected *p*-values, although corrected *p*-values were available in the CONN toolbox. The cluster-level p-uncorrected value (SPC mass/intensity) was utilized to quantify the extent and intensity of the identified clusters, offering insights into the strength of the functional connectivity patterns. We focus on identifying the cluster of interests that indicates aberrant functional connectivity between AD and HC, along with HC and MCI. Once these clusters are identified, we conduct group-wise comparisons using analysis of variance (ANOVA) among AD, HC, and MCI, with a *p*-value of 0.05.

## 4. Results

The group difference statistics of the clusters containing the connectivity of ROIs are presented in Table 2 and Table 3, and graphically in Figure 2. The identified clusters in each group difference demonstrated how disruptions in connectivity occur in AD and MCI subjects. In the AD versus HC test, cluster 1 revealed that the majority of disruptions occurred in the SMN, DAN, language, SAN and frontoparietal networks. This cluster includes the primary motor cortex, primary somatosensory cortex, and supplementary motor area contained in SMN, the intraparietal sulcus and superior parietal lobule contained in DAN, the anterior insula and anterior cingulate cortex contained in SAN, and the dorsolateral prefrontal cortex and posterior parietal cortex in frontoparietal networks. Cluster 2 includes the same anatomical regions as cluster 1 except for regions included in the frontoparietal networks. It is primarily composed of the dorsolateral prefrontal cortex and the posterior parietal cortex, including the intraparietal sulcus. Cluster 3 consists of the medial prefrontal cortex, posterior cingulate cortex, inferior parietal lobule, and lateral temporal cortex included in DMN and other anatomical regions included in SMN and DAN. In the connectome ring visualization, the connections between brain regions are color-coded based on their T-values, ranging from −3.02 indicating a decrease in connectivity in the first group compared to the second to +3.02 indicating an increase as shown in Figure 2a.

Similarly, in MCI vs. HC, cluster 1 showed disruptions in the cerebellar, visual, DAN, DMN, and SMN networks. This cluster includes the occipital lobe, including the primary visual cortex and higher-order visual areas contained in the visual network, and the cerebellar network is anchored in the dorsolateral prefrontal cortex (DLPFC) and posterior parietal cortex (PPC). Cluster 2 showed disruptions in the cerebellar network, and SMN, SAN, and DMN. In this group comparison, the T-values range from −3.19 to +3.19 as shown in Figure 2b.

Additionally, performing the ANOVA test, we found significant disruptions in all clusters in each pair of groups. The tests were conducted using *t*-tests between each pair of groups. Figure 3 shows the mean Z-score difference of the clusters between subject groups. COI1, COI2, and COI3 represent the cluster from group difference between AD and HC. Similarly, COI4 and COI5 represent the group difference between HC and MCI. *p*-values of all tests are less than 0.005, which signifies that the clusters between groups are significantly different, which is generally correct.

Additionally, we examined the correlations between the mean Z-scores of identified clusters from various networks and the global clinical dementia rating (CDR). Following best practice, we report only those results with significant correlations (*p* < 0.05) and an R value exceeding 0.35 (equivalent to R^2^ > 0.12), including associated R^2^ and *p*-values as shown in Figure 4 and Figure 5. For AD subjects, Cluster 2, encompassing the sensorimotor (SMN), dorsal attention (DAN), and salience networks, showed a significant negative correlation with CDR (R = −0.08, *p* < 0.05). Cluster 3, consisting of the default mode network (DMN), sensorimotor (SMN) network, and dorsal attention (DAN) network, displayed a significant positive correlation with CDR (R = 0.05, *p* < 0.05). Clusters 1, 4, and 5 did not achieve the required significance threshold and thus are not detailed here. For MCI subjects, Cluster 4 exhibited a significant positive correlation with CDR (R = 0.18, *p* < 0.05). Clusters 2, 3, and 5 displayed significant negative correlations with CDR, with R values of −0.13, −0.15, and −0.20, respectively (all *p* < 0.05). Though Cluster 1 did not reach the *p* < 0.05 significance threshold, it showed a positive correlation with CDR (R = 0.08).

## 5. Discussion

The major finding of this study is the identification of clusters of disrupted networks without using a statistical hard threshold. These networks consist of different ROIs that have the potential to serve as biomarkers for distinguishing AD and MCI from HC. Several studies have previously been conducted using different statistical methods to identify group differences. Consistent with these studies, we found that connections in the sensory motor network (SMN), dorsal attention network (DAN), salience network (SAN), and cerebral network were severely affected [31]. The sensory motor network includes the motor cortex and supplementary motor area. Similarly, consistent with previous studies, we discovered disruption in the default mode network (DMN) [32]. The DMN is a widely recognized large-scale brain network that encompasses various high-level cognitive regions, including the medial prefrontal cortex (mPFC), posterior cingulate cortex (PCC), and parietal regions (PTL). The DMN is commonly referred to as the “task negative” network, as its constituent regions exhibit robustly correlated activity during periods of rest and are typically deactivated during cognitive tasks that require goal-directed mental effort. The DMN is involved in episodic memory processing [33]. The findings of this study support the notion that the DMN is commonly disrupted in AD and MCI.

The sensorimotor network is associated with tasks such as converting stimulus to neuronal impulses that move throughout the brain network. The sensing process involves other networks, such as the auditory subnetwork, visual system network, salience network, dorsal attention, and DMN. As demonstrated by previous studies, this work also found disruption in connectivity in the sensory motor network in AD and MCI. Table 2 and Table 3 detail the reduced connectivity between the sensory motor network and other networks. The SAN plays a crucial role in continuously monitoring the external environment and making strategic decisions regarding the response of other brain networks to incoming information and stimuli. The SAN is responsible for regulating the transition between internal and external processing within the two primary control networks of the brain: the DMN and CEN. The SAN connectivity pattern has been reported to successfully predict different dementia types, such that decreased connectivity in the SAN correlated with behavioral variant frontotemporal dementia, whereas increased connectivity was observed in AD [34]. The current study found that the connections between the SAN and the SMN, DAN, and DMN networks are disrupted. The DAN is involved in human attention, which is perhaps the highest-level cognitive process [35]. With the disruption in DAN, a network associated with attention, there may be difficulty in focusing on goal-driven attention orientation processes. The current study identified alterations in the functional connectivity of DAN with DMN, SAN, and SMN in AD and MCI.

Additionally, the result of performing an ANOVA on three different clusters of three groups (AD, HC, and MCI) and obtaining a *p*-value of less than 0.005 indicates that there is strong evidence of a significant difference between at least one of the groups in each of the three clusters analyzed. This suggests that the groups differ significantly in at least one of the clusters, and this difference is not likely to be due to chance alone. However, without further analysis, it is difficult to determine which groups are responsible for the significant differences observed. It is important to consider the effect size and sample size of the study when interpreting the results. Post hoc tests can be conducted to determine which groups differ significantly from each other in each of the clusters, and this can provide more detailed information about the nature of the differences observed.

The main contribution of this work is the development and application of a novel method for identifying clusters of disrupted networks in AD and MCI without relying on a statistical hard threshold. This method allows for the characterization of specific ROIs within different brain networks that have the potential to serve as biomarkers for distinguishing AD and MCI from HC. We further assessed the correlations between the mean Z-scores of identified clusters from these networks and the global CDR. For AD subjects, correlations showed significant findings for clusters related to the SMN, DAN, and SAN, as well as the DMN. For MCI subjects, significant correlations spanned across multiple clusters with varying positive and negative associations with CDR.

The study highlights several key findings:

Disrupted Network Clusters: The study identifies disrupted network clusters encompassing different brain networks, including the SMN, DAN, SAN and cerebral network. These findings provide insights into the specific regions and connections that are severely affected in AD and MCI.

Default Mode Network (DMN) Disruption: The study reinforces previous research by confirming disruption in the DMN, a well-known large-scale brain network associated with high-level cognitive functions. This disruption has implications for episodic memory processing and cognitive deficits observed in AD and MCI.

Biomarker Potential: The identified disrupted network clusters, particularly within the DMN and other associated networks, hold the potential to serve as biomarkers for distinguishing AD and MCI from healthy controls. This suggests the feasibility of utilizing specific brain network disruptions as diagnostic markers for neurodegenerative diseases.

Support for Existing Theories: By corroborating findings from previous studies that have implicated the SMN, DAN, SAN, and DMN in AD and MCI, the study strengthens existing theoretical frameworks that link these networks to cognitive decline and neurodegeneration.

Limitations: The study’s primary limitation is its small sample size. The dataset used for analysis includes only 33 AD, 31 MCI, and 31 HC subjects from the ADNI2 cohort. This limited sample size may not be representative of the broader population and could lead to issues with generalizability. While the integration of both structural sMRI and fMRI data is beneficial for better registration and alignment, this approach resulted in a reduction in the number of available samples for analysis. The study’s findings might have been influenced by the availability of both modalities within the ADNI2 cohort. While the study introduces a novel method for identifying disrupted networks without a statistical hard threshold, the details about this method and its potential limitations are not fully described. The lack of a statistical threshold might raise questions about the reliability and reproducibility of the results. The study appears to have a cross-sectional design, which limits the ability to establish causal relationships between disrupted networks and AD or MCI. Longitudinal studies are needed to better understand the progression of these disruptions over time. While the study suggests that the identified disrupted networks could serve as potential biomarkers for distinguishing AD and MCI from healthy controls, further validation is necessary. Validation studies should involve larger and more diverse samples and employ rigorous statistical methods to assess the diagnostic accuracy of these biomarkers.

Future Work: Longitudinal Studies: Conducting longitudinal studies that follow participants over time could provide insights into the temporal dynamics of disrupted networks in AD and MCI. This would help determine whether these disruptions are causal factors or consequences of the disease progression. Replicating the study with larger and more diverse cohorts would enhance the generalizability of the findings and strengthen the validity of the identified disrupted networks as potential biomarkers. Further development and refinement of the novel method for identifying disrupted networks without a statistical hard threshold are warranted. Comparative analyses with existing statistical approaches could help establish the reliability and validity of this method. Given the potential benefits of integrating both structural and functional imaging data, future studies could explore advanced techniques for multimodal data integration, which might improve the accuracy and robustness of identified biomarkers. To establish causal relationships between disrupted networks and AD/MCI, experimental designs such as causal inference methods or interventions targeting these networks could be considered. Future research could focus on translating the identified disrupted networks into clinical practice. Developing tools or algorithms that utilize these biomarkers for early diagnosis, disease monitoring, or treatment evaluation could have significant clinical implications. Leveraging advanced machine learning and artificial intelligence techniques could aid in the identification and validation of biomarkers from complex brain imaging data.

Clinical Usage:

Early Diagnosis and Prognosis: The identified disrupted network clusters, particularly within the DMN and other associated networks, could potentially serve as biomarkers for early diagnosis and prognosis of AD and MCI. Clinicians could use these biomarkers to identify individuals at risk of developing AD or track the progression of cognitive decline.

Treatment Monitoring: The disrupted network clusters could be employed to monitor the effects of therapeutic interventions for AD and MCI. Changes in these biomarkers over time could help assess the efficacy of treatments and guide treatment adjustments.

Personalized Treatment Approaches: By identifying specific disrupted networks in individual patients, clinicians could tailor treatment strategies to target the affected brain regions, potentially leading to more personalized and effective interventions.

Clinical Trials: The identified biomarkers could be valuable in clinical trial design by aiding in patient selection, monitoring treatment effects, and assessing the impact of interventions on the disrupted networks.

Disease Subtyping: The distinct patterns of disrupted networks in different subtypes of AD or MCI could contribute to more precise disease subtyping, leading to improved diagnostic accuracy and tailored therapeutic approaches.

Computational Applicability:

Machine Learning and Artificial Intelligence: The methodological innovation of identifying disrupted network clusters without a statistical hard threshold could be integrated into machine learning and artificial intelligence algorithms. These algorithms could enhance the automated detection and classification of AD and MCI based on brain network disruptions.

Predictive Models: The identified disrupted network clusters could be incorporated into predictive models that utilize multimodal imaging data to forecast an individual’s risk of developing AD or MCI, providing insights for early intervention and disease management.

Data-Driven Biomarker Development: The computational approach could facilitate the discovery of new imaging-based biomarkers beyond the ones identified in this study. By leveraging large datasets and advanced analytical techniques, researchers could uncover additional network disruptions associated with AD and MCI.

Data Integration: The integration of structural MRI (sMRI) and functional MRI (fMRI) data, as demonstrated in this study, could become a standard approach in neuroimaging research, leading to improved spatial localization and more accurate identification of disrupted brain networks

## 6. Conclusions

This study utilized a threshold-free network-based statistics approach to identify clusters of disrupted networks in AD and MCI patients compared to healthy controls. The identified networks, including the sensory motor network, default mode network, and salience network have the potential to serve as biomarkers for distinguishing AD and MCI from HC. The study underscores the crucial role of these networks in cognitive function and their disruption in neurodegenerative conditions. The findings of this study support previous research showing disruptions in these networks and highlight the importance of studying network-level changes in neurodegenerative diseases. The identification of these disrupted networks could lead to a better understanding of the underlying pathophysiology of AD and MCI, as well as the development of novel therapeutic targets. Overall, this study provides valuable insights into the functional connectivity changes that occur in AD and MCI and underscores the importance of investigating network-level alterations in neurodegenerative diseases.

## Figures and Tables

**Figure 1 diagnostics-13-03074-f001:**
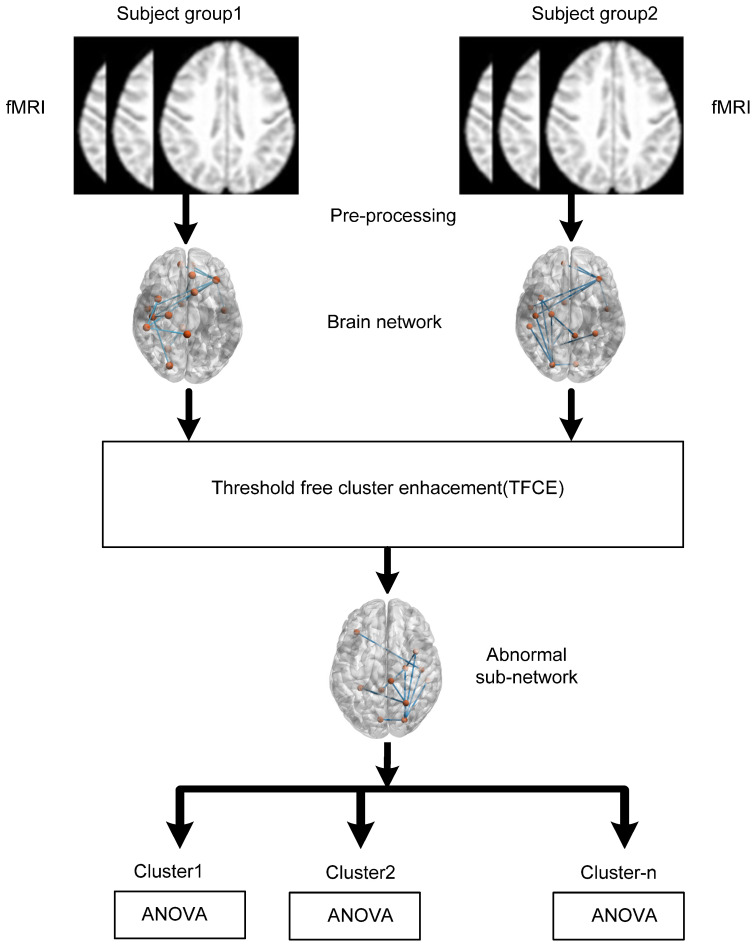
Diagram illustrating the sequence of data analysis steps.

**Figure 2 diagnostics-13-03074-f002:**
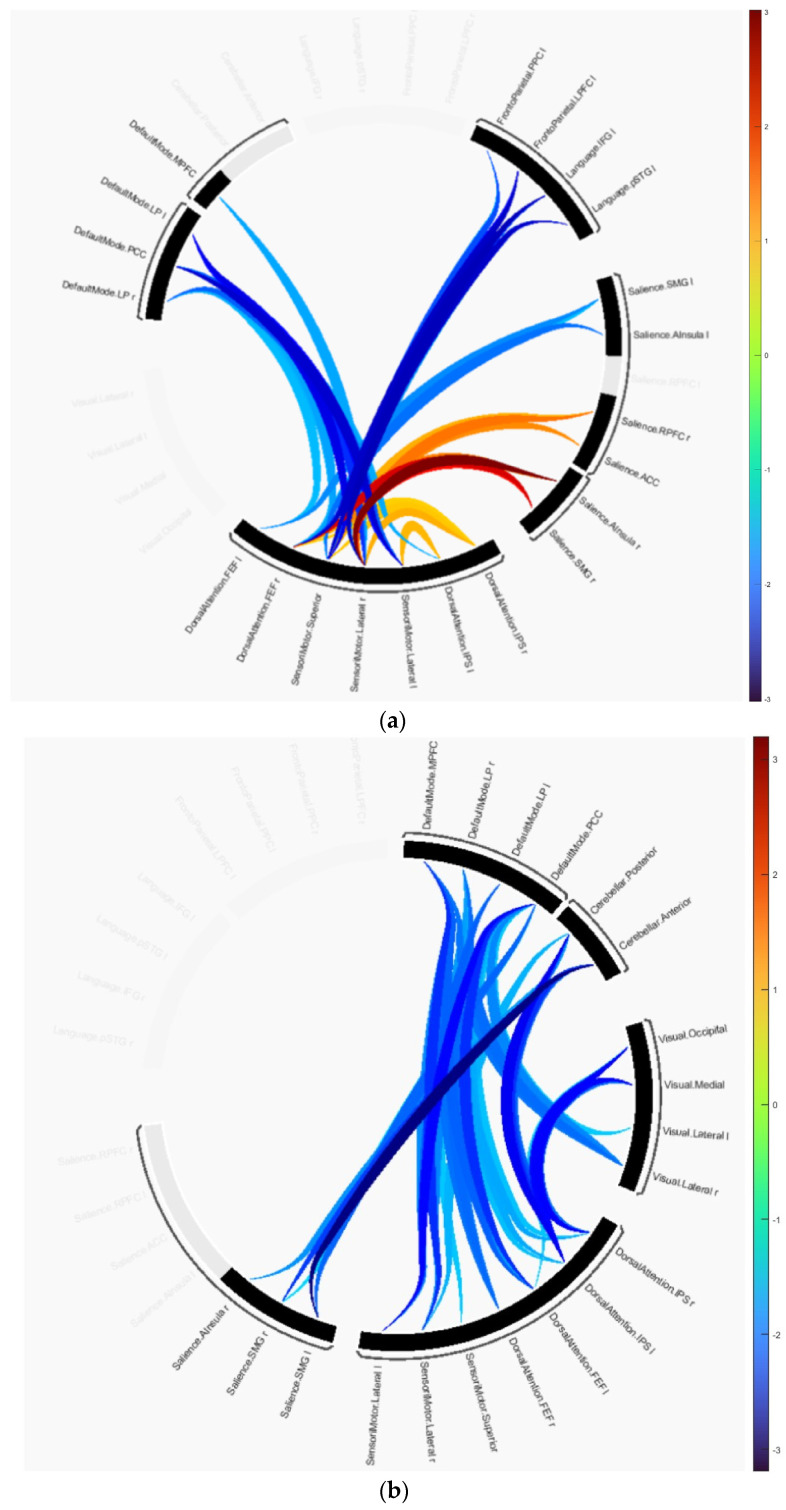
Group differences represented as connectome rings: (**a**) HC versus AD, (**b**) HC versus MCI.

**Figure 3 diagnostics-13-03074-f003:**
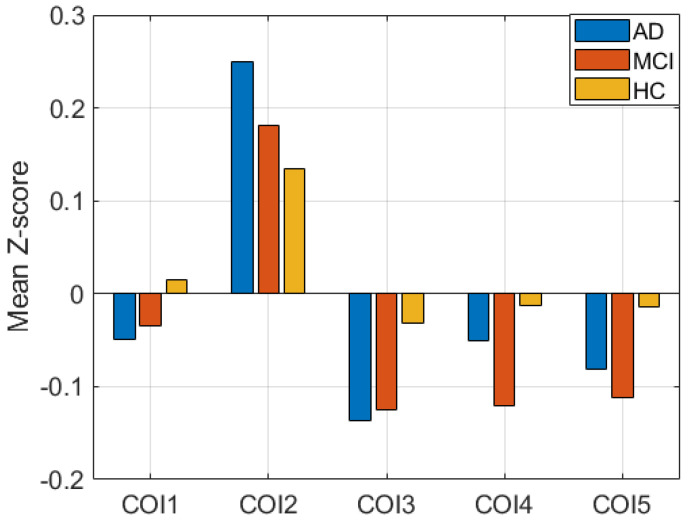
Mean Z-score difference between three groups in five clusters of interest (COIs).

**Figure 4 diagnostics-13-03074-f004:**
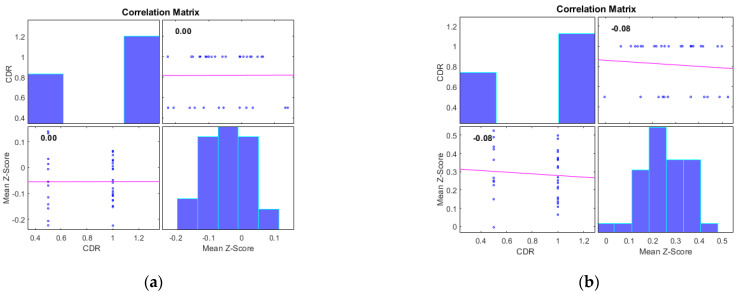
Scatter plots illustrating the correlation between mean Z-scores of functional connectivity and global clinical dementia rating (CDR) in AD subjects for (**a**) Cluster1, (**b**) Cluster2, (**c**) Cluster3, (**d**) Cluster4, and (**e**) Cluster5.

**Figure 5 diagnostics-13-03074-f005:**
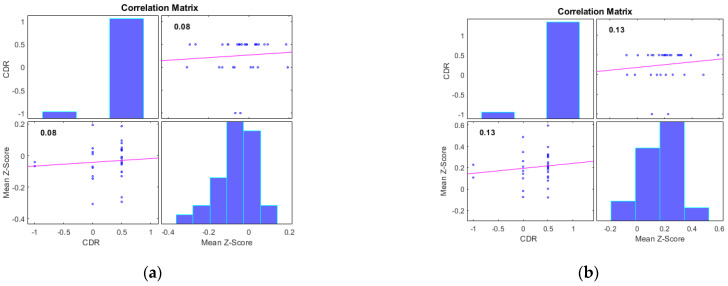
Scatter plots illustrating the correlation between mean Z-scores of functional connectivity and global clinical dementia rating (CDR) in MCI subjects for (**a**) Cluster1, (**b**) Cluster2, (**c**) Cluster3, (**d**) Cluster4, and (**e**) Cluster5.

**Table 1 diagnostics-13-03074-t001:** Subject demographic status.

Number of Subjects	HC (*n* = 31)	MCI (*n* = 31)	AD (*n* = 33)
Age (years)	73.9 ± 5.4	74.5 ± 5.0	72.7 ± 7.0
Global CDR	0.04 ± 0.13	0.5 ± 0.18	0.95 ± 0.30
MMSE	28.9 ± 1.65	27.5 ± 2.02	20.87 ± 3.6

**Table 2 diagnostics-13-03074-t002:** Statistical analysis of connectivity differences in AD and HC using TFCE.

Cluster 1		Statistic
SensoriMotor.Lateral (R)	Language.IFG (L)	p-uncorrected = 0.022849
SensoriMotor.Superior	Language.pSTG (L)	Mass = 105.17
DorsalAttention.FEF (R)	FrontoParietal.LPFC (L)	
SensoriMotor.Lateral (R)	FrontoParietal.LPFC (L)	
Language.IFG (L)	SensoriMotor.Superior	
DorsalAttention.FEF (R)	Language.pSTG (L)	
DorsalAttention.FEF (R)	Language.IFG (L)	
SensoriMotor.Superior	FrontoParietal.PPC (L)	
Salience.AInsula (L)	SensoriMotor.Superior	
FrontoParietal.LPFC (L)	SensoriMotor.Superior	
Salience.SMG (L)	DorsalAttention.FEF (L)	
SensoriMotor.Lateral (R)	Language.pSTG (L)	
SensoriMotor.Lateral (R)	FrontoParietal.PPC (L)	
Salience.SMG (L)	SensoriMotor.Superior	
**Cluster 2**		**Statistic**
SensoriMotor.Lateral (R)	Salience.AInsula (R)	p-uncorrected = 0.03837
DorsalAttention.FEF (R)	Salience.SMG (R)	Mass = 74.73
Salience.SMG (R)	SensoriMotor.Lateral (R)	
SensoriMotor.Lateral (R)	Salience.RPFC (R)	
DorsalAttention.FEF (R)	Salience.AInsula (R)	
Salience.SMG (R)	SensoriMotor.Superior	
Salience.ACC	SensoriMotor.Lateral (R)	
Salience.RPFC (R)	SensoriMotor.Superior	
SensoriMotor.Lateral (R)	DorsalAttention.IPS (R)	
DorsalAttention.FEF (R)	Salience.RPFC (R)	
DorsalAttention.IPS (L)	SensoriMotor.Lateral (L)	
DorsalAttention.IPS (R)	SensoriMotor.Superior	
**Cluster 3**		**Statistic**
SensoriMotor.Lateral (L)	DefaultMode.PCC	p-uncorrected = 0.04207
DefaultMode.LP (L)	SensoriMotor.Superior	Mass = 70.06
SensoriMotor.Lateral (L)	DefaultMode.LP (L)	
SensoriMotor.Lateral (R)	DefaultMode.LP (L)	
SensoriMotor.Lateral (R)	DefaultMode.PCC	
DefaultMode.LP (R)	SensoriMotor.Lateral (L)	
DefaultMode.LP (L)	DorsalAttention.FEF (R)	
SensoriMotor.Lateral (R)	DefaultMode.MPFC	
DefaultMode.LP (R)	DorsalAttention.IPS (L)	
DorsalAttention.FEF (R)	DefaultMode.PCC	
DefaultMode.LP (R)	DorsalAttention.FEF (L)	

**Table 3 diagnostics-13-03074-t003:** Statistical analysis of connectivity differences in MCI and HC using TFCE.

Cluster 1		Statistic
Cerebellar.Posterior	DorsalAttention.IPS (L)	p-uncorrected = 0.016848
Visual.Occipital	DorsalAttention.IPS (R)	Mass = 160.07
Visual.Occipital	DorsalAttention.IPS (L)	
DorsalAttention.FEF (L)	DefaultMode.PCC	
SensoriMotor.Lateral (R)	DefaultMode.MPFC	
Cerebellar.Anterior	DorsalAttention.IPS (L)	
Cerebellar.Posterior	DorsalAttention.IPS (R)	
DefaultMode.MPFC	DorsalAttention.FEF (L)	
DefaultMode.LP (L)	DorsalAttention.FEF (R)	
Cerebellar.Anterion	DorsalAttention.IPS (R)	
DefaultMode.LP (R)	Visual.Lateral (R)	
Visual.Medial	DorsalAttention.IPS (R)	
DorsalAttention.FEF (R)	DefaultMode.MPFC	
Visual.Lateral (R)	DefaultMode.MPFC	
DorsalAttention.FEF (R)	DefaultMode.PCC	
Visual.Lateral (R)	DefaultMode.PCC	
DefaultMode.MPFC	SensoriMotor.Lateral (L)	
DefaultMode.LP (R)	DorsalAttention.FEF (L)	
DefaultMode.LP (R)	DorsalAttention.IPS (R)	
DorsalAttention.IPS (L)	DefaultMode.PCC	
DefaultMode.LP (R)	DorsalAttention.IPS (L)	
DorsalAttention.IPS (R)	DefaultMode.PCC	
DefaultMode.MPFC	SensoriMotor.Superior	
Visual.Occipital	DorsalAttention.FEF (L)	
Visual.Lateral (L)	DefaultMode.PCC	
**Cluster 2**		**Statistic**
SensoriMotor.Lateral (L)	DefaultMode.PCC	p-uncorrected = 0.047229
Cerebellar.Anterior	Salience.SMG (R)	Mass = 77.93
SensoriMotor.Lateral (L)	DefaultMode.LP (L)	
Salience.SMG (L)	DefaultMode.PCC	
SensoriMotor.Lateral (R)	DefaultMode.PCC	
Cerebellar.Anterior	Salience.AInsula (R)	
SensoriMotor.Lateral (R)	DefaultMode.LP (L)	
Cerebellar.Posterior	Salience.AInsula (R)	
Cerebellar.Posterior	SensoriMotor.Lateral (R)	
Cerebellar.Posterior	Salience.SMG (L)	
Salience.SMG (R)	DefaultMode.PCC	

**Acronyms:** MPFC: medial prefrontal cortex, LP: lateral posterior, IPS: intraparietal sulcus, PCC: posterior cingulate cortex, FEF: frontal eye field, SMG: supramarginal gyrus, pSTG: posterior superior temporal gyrus, IFG: inferior frontal gyrus, LPFC: lateral prefrontal cortex, PPC: posterior parietal cortex, RPFC: rostral prefrontal cortex.

## Data Availability

The datasets used in this study were obtained from the ADNI webpage, which is freely accessible for all scientists and investigators to conduct experiments on Alzheimer’s disease and can be accessed from ADNI’s website: http://adni.loni.usc.edu/about/contact-us/, accessed on 27 October 2021. The raw data backing the results of this research will be made accessible by the authors, without undue reservation.

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
