# Peer review of "Resting-State Functional Connectivity Difference in Alzheimer’s Disease and Mild Cognitive Impairment Using Threshold-Free Cluster Enhancement"

_diagnostics, 2023, doi:10.3390/diagnostics13193074_

Round 1
Reviewer 1 Report
The authors have investigated whether functional connectivity in large-scale networks is specifically altered in AD and MCI patients compared to HC.
First of all, references must be revised because only few of them (maybe 5?) are more recent than 2018 and only few works dealing with resting-state networks further than DMN were cited. E.g., look at 10.1089/brain.2022.0032, https://doi.org/10.3389/fnagi.2023.1204134 and 10.3389/fnins.2014.00223. Also, the total number of references is scarce and the references must be cited sequentially through the manuscript (e.g., ref. 27 comes before 21).
The introduction extensively reports findings from previous literature but the aim of the present work is not well described. It seems that the aim was "investigate brain connectivity" without any hypothesis to verify or any specific goal.
The method section could be partially simplified by removing some repeating parts. Further, the brain parcellation description is completely missing: the authors must add which atlases were used to parcellate the brain and which specific ROIs and networks were extracted for the analysis. Why only 32 ROIs were selected between the 164 identified?
Discussion is sometimes hard to follow because the figures lack of legends and appropriate captions.
As a general comment, all abbreviations and acronyms should be defined and re-checked (e.g. HC and NC), also there are some typos (e.g. row 287).
Detailed comments:
Introduction:
- what do you mean with TFNBS? To my knowledge NBS has two advanced statistical methods, which are NBS and FDR (not TFCE). The standard TFCE approach I am aware of is the one used by FSL randomise to correct the statistical threshold taking into account the cluster extent. Also your references are in line with knowledge, so please clarify this point.
Methods:
- Subjects: Please add the gender distribution within AD, MCI and HC groups. You must also describe which MRI sequences were used: What do you mean with "MRI and fMRI data"? What does MRI stand for? structural 3DT1 weighted images?
- Please change MCI space to MNI space
- Data preprocessing: Please be clear with the pipeline and don't repeat several times the same steps. This section could be simplified by removing these repeating parts.
- Brain Connectivity Estimation: add parcellation description and explain why only 32 ROIs were selected.
- TFNBS: here you explained the different between standard NBS and TFNBS, but actually is not clear if you developed the second method (if so you MUST say this!) or if you use it (references are missing). In any case, if you are not using NBS tool you should say the software that you used.
Results:
- Please clarify which anatomical regions are included in which clusters. Table 2 and 3 report the labels of the affected network/regions but you must define all the acronyms. Also tables should be of clearer reading if you "cluster" all regions belonging to the same network. Modify Figure 2 accordingly to the comment below.
- You used the term "disrupted" to indicate pathological alterations, but since all legends are missing is hard to follow these sentences. Does it mean that "disrupted" is equal to reduced? Are there any "compensation" in AD and MCI? Please clarify these findings.
- It is not clear why you looked at uncorrected p-values since you used an "advanced" statistical approach. Did you try using standard NBS but with corrected p-values? Something changes?
- This sentence seems to be wrong: "P-values of all tests are less 239 than 0.005, which signifies that "the clusters between groups are significantly different is 240 generally correct. " because all values in Tables 2/3 are slightly lees than 0.05.
- Please report only significant correlations, non-significant ones cannot be correctly interpreted. Surely correlations with r=0.08 cannot be significant.
Discussion:
- To obtain direct differences between pairs of groups, you should perform a pair-wise post-hoc comparison after the ANOVA test (as also you mentioned). In this way it would be easier to discuss results like in the paragraph from 296-303.
- In the conclusion (row 339) you mentioned that the cerebellar network as long with DMN has the potential to serve as biomarker, but you never mentioned before this cerebellar network. Thus, what about SAN or SMN? Please clarify and justify your conclusions.
Figures:
- In all figures you must add the legend of the colours, the panel letters when needed, and modify accordingly the caption.
- Fig 2: Indicate if the mean difference is significant.
Author Response
Reviewer 1:
Thank you for your review and for giving helpful comments which were very useful in revising the paper. According to your comments, we have revised the paper as follows:
First of all, references must be revised because only few of them (maybe 5?) are more recent than 2018 and only few works dealing with resting-state networks further than DMN were cited. E.g., look at 10.1089/brain.2022.0032, https://doi.org/10.3389/fnagi.2023.1204134 and 10.3389/fnins.2014.00223. Also, the total number of references is scarce and the references must be cited sequentially through the manuscript (e.g., ref. 27 comes before 21).
As suggested by reviewer we added recently published papers in revised manuscript and order of citation is also rearranged in revised manuscript.
” Another study used a computational brain network model to link neuronal hyperactivity to large-scale oscillatory slowing in early-stage Alzheimer's disease [17]. A machine learning study found altered large-scale dynamic connectivity patterns in AD and MCI patients [18]. Additionally, a triple-network dynamic connection study demonstrated that AD is associated with abnormal organization and function of large-scale brain networks [19]. Therefore, we included other networks in our investigation of brain connectivity disruptions. Most studies are performed using threshold-based statistical methods to generate edgewise significance values by defining an a priori clustering threshold. In this study, we used threshold-free cluster enhancement method (TFCE) [21]. TFCE generates edge-wise significant values and does not require the a priori definition of an edge-defining hard threshold. Thus, TFCE provides a better insight into the altered cluster structure of brain connectivity. “
In Reference
- American Psychiatric Association and American Psychiatric Association (1994) Task Force on DSM-IV., “Diagnostic and sta-tistical manual of mental disorders,” DSM-IV, vol xxv, 4th edn. American Psychiatric Association, Washington, DC.
- Alzheimer’s association, “2023 Alzheimer's disease facts and figures,” available at https://www.alz.org/media/documents/alzheimers-facts-and-figures.pdf
- Guzmán-Vélez, E., Diez, I., Schoemaker, D., Pardilla-Delgado, E., Vila-Castelar, C., Fox-Fuller, J. T., ... & Quiroz, Y. T. (2022). Amyloid-β and tau pathologies relate to distinctive brain dysconnectomics in preclinical autosomal-dominant Alzheimer’s disease. Proceedings of the National Academy of Sciences, 119(15), e2113641119.
- Zhang, Y., Chen, H., Li, R., Sterling, K., & Song, W. (2023). Amyloid β-based therapy for Alzheimer’s disease: challenges, successes and future. Signal transduction and targeted therapy, 8(1), 2
- DeTure, M. A., & Dickson, D. W. (2019). The neuropathological diagnosis of Alzheimer’s disease. Molecular neurodegenera-tion, 14(1), 1-18.
- V. Menon, “Large-scale brain networks and psychopathology: A unifying triple network model,” Trends in Cognitive Sci-ences, vol. 15, no. 10. pp. 483–506, Oct. 2011. doi: 10.1016/j.tics.2011.08.003.
- R. K. Lama and G. R. Kwon, “Diagnosis of Alzheimer’s Disease Using Brain Network,” Frontiers in Neuroscience, vol. 15, Feb. 2021, doi: 10.3389/fnins.2021.605115.
- Won, K.A. Nielson, and J. C. Smith, (2023). Large-Scale Network Connectivity and Cognitive Function Changes After Exercise Training in Older Adults with Intact Cognition and Mild Cognitive Impairment. Journal of Alzheimer's Disease Reports, (Preprint), 1-15.
- van Nifterick, A. A. Gouw, R. E. van Kesteren, P. Scheltens, C. J. Stam, C. J., and W. de Haan, (2022). A multiscale brain network model links Alzheimer’s disease-mediated neuronal hyperactivity to large-scale oscillatory slowing. Alzheimer's Research & Therapy, 14(1), 1-20.
- Jing, P. Chen, Y. Wei, J. Si, Y. Zhou, D. Wang, D., and Alzheimer's Disease Neuroimaging Initiative. (2023). Altered large‐scale dynamic connectivity patterns in Alzheimer's disease and mild cognitive impairment patients: A machine learning study. Human Brain Mapping.
- Meng, Y. Wu, Y. Liang, D. Zhang, Z. Xu, X. Yang, and L. Meng, L. (2022). A triple-network dynamic connection study in Alzheimer's disease. Frontiers in psychiatry, 13, 862958.
The introduction extensively reports findings from previous literature but the aim of the present work is not well described. It seems that the aim was "investigate brain connectivity" without any hypothesis to verify or any specific goal.
As suggested by reviewer we further elaborated the hypothesis statement in revised manuscript as “We hypothesize that individuals with AD and MCI exhibit disrupted functional connectivity within key brain networks, including the SMN, DAN, SAN, and DMN. This disruption, characterized by reduced interregional correlations and connectivity patterns, may serve as a potential biomarker for distinguishing AD and MCI from healthy controls. The hypothesis further posits that the observed impairments within the DMN, a network associated with episodic memory processing and cognitive rest, contribute significantly to the cognitive deficits observed in AD and MCI individuals. To test this hypothesis, we di-vided the whole brain into 164 different anatomical regions of interest (ROIs) and used eight important large-scale networks comprising only 32 ROIs.”
(Discussion is sometimes hard to follow because the figures lack of legends and appropriate captions.
As a general comment, all abbreviations and acronyms should be defined and re-checked (e.g. HC and NC), also there are some typos (e.g. row 287)..
As suggested by reviewer all abbreviations and acronyms are rechecked in revised manuscript
Detailed comments:
Introduction:
- what do you mean with TFNBS? To my knowledge NBS has two advanced statistical methods, which are NBS and FDR (not TFCE). The standard TFCE approach I am aware of is the one used by FSL randomise to correct the statistical threshold taking into account the cluster extent. Also your references are in line with knowledge, so please clarify this point..
As suggested by reviewer to make clarity on presentation, we made update on introduction section with “Network-based statistics (NBS) [18] [19] is a parameter-free statistical tool used to identify group differences when the distribution is known. The first step of NBS is to estimate the connectivity matrix r, which is of size N×N. Each element of this matrix, r(i, j) , represents the edge value between two regions of interest (ROIs) i and j. After estimating the connectivity matrix, the second step of NBS is to apply a threshold on the edge values r(i, j) to define a cluster. Next, the thresholded connections are grouped for each subject belonging to the defined group. Finally, a permutation test is performed on these edge values, repeating the process for n different combinations. We count the number of test statistics whose value is greater than the initial test statistics, and the p-value is calculated by estimating the ratio of this number to the total number of test statistics.
In contrast, threshold-free cluster (TFCE) performs an enhancement operation for each edge value r(i, j) instead of a thresholding operation. We start with a ROI-to-ROI connectivity matrix, which is derived from a General Linear Model analysis. ROIs are sorted automatically using a hierarchical clustering procedure, which considers anatomical proximity or functional similarity metrics. Instead of using a fixed height threshold as in NBS, TFCE computes a TFCE score map [16] [26]. The TFCE score map combines the strength of the statistical effect for each connection with the extent of neighboring connections that show similar effects.
(2)
where represents the extension of neighboring connections and represents the height of the fractional edge value. and are extension and height enhancement parameters, respectively. This helps capture both local and distributed effects. The expected distribution of TFCE values under the null hypothesis is estimated using permutation iterations of the original data using 1,000 iterations. For each cluster in the original analysis, a peak-level Family-Wise Error (FWE)-corrected p-value is computed. This indicates the likelihood under the null hypothesis of observing at least one or more connections with the given TFCE scores over the entire ROI-to-ROI connectivity matrix. For the peak-level analysis, local peaks in the TFCE score map are compared to the null hypothesis distribution of local-peak TFCE values. This estimation provides uncorrected p-values for each peak, representing the likelihood under the null hypothesis of observing a peak with similar or larger scores by chance.
Methods:
- Subjects: Please add the gender distribution within AD, MCI and HC groups. You must also describe which MRI sequences were used: What do you mean with "MRI and fMRI data"? What does MRI stand for? structural 3DT1 weighted images?
As suggested by reviewer we added gender information in revised manuscript as “Subjects with the following demographic status were selected in our study out of all available data in the ADNI database. HC group consists 31 subjects with 14 males and 17 females, MCI consists 17 males and 14 females, and HC group consists 15 males and 18 females.”
As suggested by reviewer we added additional information of MR images “We selected the subjects according to the availability of both structural MRI (sMRI) and fMRI imaging data. sMRI is a T1-weighted sequence that provides high-resolution anaomical images of the brain. Subjects with the following demographic status were selected in our study out of all available data in the ADNI database:”
Please change MCI space to MNI space
AS suggested by reviewer we changed the MCI space to MCI space in revised manuscript”
- Data preprocessing: Please be clear with the pipeline and don't repeat several times the same steps. This section could be simplified by removing these repeating parts.
As suggested by reviewer, we reviewed the steps carefully and deleted some repeated steps.
- Brain Connectivity Estimation: add parcellation description and explain why only 32 ROIs were selected.
We did not find a specific process as parcellation however the segmentation process in CONN tool box is described as “This was done by segmenting the functional and anatomical data into gray matter, white matter, and cerebrospinal fluid (CSF) classes using the SPM12 unified segmentation method [29]. Outlier detection was performed after normalization and segmentation. The difference image of mean BOLD signal was used as a reference image for the functional data, while the T1-weighted volume was used as a reference image for the structural data [29]-[30].”
The reason for selecting 32 ROIs is described as “The functional connectivity among different ROIs was estimated on the basis of temporal correlations of BOLD signals in these regions. Initially, we extracted the BOLD time series from 164 regions. Among these regions, only 32 ROIs were selected to construct 8 important large-scale networks: default mode (4 ROIs), sensorimotor (3 ROIs), visual (4 ROIs), salience/cingulo-opercular (7 ROIs), dorsal attention (4 ROIs), frontoparietal/central executive (4 ROIs), language (4 ROIs) and cerebellar (2 ROIs). Consistent with previous studies, Pearson’s correlation was estimated among the pairs of regions and expressed in terms of an ROI to ROI correlation matrix.”
TFNBS: here you explained the different between standard NBS and TFNBS, but actually is not clear if you developed the second method (if so you MUST say this!) or if you use it (references are missing). In any case, if you are not using NBS tool you should say the software that you used.
In this study, we used the TFCE of CONN tool-box. Since the TFCE and TFNBS works in similar manner, for the simplicity of explanation we used the term TFNBS. However, as per reviewer’s opinion the TFNBS created more confusion. So, we included only the TFCE of CONN toolbox with all the settings used during the experiments. The Result section of revised manuscript contains the details of TFCE settings.
Results:
We implemented the TFCE method of the CONN toolbox to identify clusters of connections that are statistically significant, interpreted in terms of functional connectivity. The statistical parameters utilized in the CONN toolbox for functional connectivity analysis were configured as follows: for the connection thresholding, connection threshold greater than 0.05 was employed to ensure that only robust connections were considered for further analysis. The statistical mode was set to F/T/X, which enables assessment using various statistical metrics, enhancing the reliability of identified connections. For the cluster thresholding, to detect significant clusters of connected brain regions, a cluster threshold (P) of less than 0.05 was applied. This approach aids in identifying groups of functionally connected regions that exhibit strong correlation patterns. The cluster-level p-uncorrected (SPC mass/intensity) was utilized to quantify the extent and intensity of the identified clusters, offering insights into the strength of the functional connectivity patterns.
Results:
- Please clarify which anatomical regions are included in which clusters. Table 2 and 3 report the labels of the affected network/regions but you must define all the acronyms. Also, tables should be of clearer reading if you "cluster" all regions belonging to the same network. Modify Figure 2 accordingly to the comment below.
As Suggested by reviewer we put the Acronym and anaotomical regions in revised manuscript. Figure e is also changed.
Acronyms: MPFC: medial prefrontal cortex, LP: lateral posterior, IPS: intraparietal sulcus, PCC: posterior cingulate cortex, FEF: frontal eye field, SMG: supramarginal gyrus, pSTG : posterior superior temporal gyrus, IFG: inferior frontal gyrus, LPFC: lateral pre-frontal cortex, PPC: posterior parietal cortex, , RPFC: rostral prefrontal cortex.
The group difference statistics of the clusters containing the connectivity of ROIs are presented in Table 2 and 3, and graphically in Figure 3. The identified clusters in each group difference demonstrated how disruptions in connectivity occur in AD and MCI subjects. In AD versus HC test, cluster 1 revealed that the majority of disruptions occurred in the SMN, DAN, Language, SAN and Frontoparietal net-works. This cluster includes the primary motor cortex, primary somatosensory cortex, and supplementary motor area contained in SMN, intraparietal sulcus and superior parietal lobule contained in DAN, anterior insula and anterior cingulate cortex contained in SAN, and Dorsolateral prefrontal cortex and posterior parietal cortex in Frontoparietal networks. Cluster2 includes same anatomical regions as cluster1 except regions included by Frontoparietal networks. It is primarily composed of the dorsolateral prefrontal cortex and the posterior parietal cortex, including the intraparietal sulcus. Cluster3 consists of Medial prefrontal cortex, posterior cingulate cortex, inferior parietal lobule, lateral temporal cortex included by DMN and other anatomical regions included by SMN, and DAN. Similarly, in MCI vs HC, cluster 1 showed disruptions in the network of Cerebellar, Visual, DAN, DMN, and SMN. This cluster includes the Occipital lobe, including the primary visual cortex and higher-order visual areas contained by visual network and Cerebellar network is anchored in the dorsolateral prefrontal cortex (DLPFC) and posterior parietal cortex (PPC). Cluster 2 showed disruptions in the Cerebellar network, SMN, SAN, and DMN.
- You used the term "disrupted" to indicate pathological alterations, but since all legends are missing is hard to follow these sentences. Does it mean that "disrupted" is equal to reduced? Are there any "compensation" in AD and MCI? Please clarify these findings.
As suggested by reviewer to make clarity on meaning of disrupted we added following sentence in Introduction section “We hypothesize that individuals with AD and MCI exhibit disrupted functional connectivity within key brain networks, including the SMN, DAN, SAN, and DMN. This disruption, characterized by reduced interregional correlations and connectivity patterns, may serve as a potential biomarker for distinguishing AD and MCI from healthy controls”
It is not clear why you looked at uncorrected p-values since you used an "advanced" statistical approach. Did you try using standard NBS but with corrected p-values? Something changes?
We implemented the TFCE method of the CONN toolbox to identify clusters of connection that are statistically significant, interpreted in terms of functional connectivity. The statistical parameters utilized in the CONN toolbox for functional connectivity analysis were configured as follows: for the connection thresholding, connection threshold greater than 0.05 was employed to ensure that only robust connections were considered for further analysis. The statistical mode was set to F/T/X, which enables assessment using various statistical metrics, enhancing the reliability of identified connections. For the cluster thresholding, to detect significant clusters of connected brain regions, a cluster threshold (P) of less than 0.05 was applied. This approach aids in identifying groups of functionally connected regions that exhibit strong correlation patterns. The cluster-level p-uncorrected (SPC mass/intensity) was utilized to quantify the extent and intensity of the identified clusters, offering insights into the strength of the functional connectivity patterns.
This sentence seems to be wrong: "P-values of all tests are less 239 than 0.005, which signifies that "the clusters between groups are significantly different is 240 generally correct. " because all values in Tables 2/3 are slightly less than 0.05.
“Additionally, performing the ANOVA test, we found significant disruptions in all clusters in each pair of groups. The tests were conducted using t-tests between each pair of groups. Figure 2 shows the mean Z-score difference of clusters between subject groups. COI1, COI2 and COI3 rep-resent the cluster from group difference between AD and HC. Similarly, COI4 and COI5 represent the group difference between HC and MCI. P-values of all tests are less than 0.005, which signifies that the clusters between groups are significantly different is generally correct.”
This paragraph describes the test result obtained after ANOVA of all clusters. The p-values obtained on this test were less than 0.005 which lesser than that we obtained in TFCE. So, this p-value and p-value from Table 2 and 3 are different.
- Please report only significant correlations, non-significant ones cannot be correctly interpreted. Surely correlations with r=0.08 cannot be significant.
As suggested by reviewer, the mistake is corrected in revised manuscript as “In MCI subjects, the correlation on these clusters was mixed. Specifically, cluster 4 (0.18) was positively correlated, while cluster2 (-0.13), cluster3 (-0.15), and cluster 5 (-0.20) were negatively correlated and cluster 1 (0.08) was nearly uncorrelated.”
Discussion:
- To obtain direct differences between pairs of groups, you should perform a pair-wise post-hoc comparison after the ANOVA test (as also you mentioned). In this way it would be easier to discuss results like in the paragraph from 296-303.
- In the conclusion (row 339) you mentioned that the cerebellar network as long with DMN has the potential to serve as biomarker, but you never mentioned before this cerebellar network. Thus, what about SAN or SMN? Please clarify and justify your conclusions.
As suggested by reviewer the modification in Conclusion is made as “This study utilized a threshold free network-based statistics approach to identify clusters of disrupted networks in AD and MCI patients compared to healthy controls. The identified networks, including the sensory motor network, sensory motor network, default mode network, and salience network have the potential to serve as biomarkers for distinguishing AD and MCI from HC. The study underscores the crucial role of these networks in cognitive function and their disruption in neurodegenerative conditions. The findings of this study support previous research showing disruptions in these networks and highlight the importance of studying network-level changes in neurodegenerative diseases. The identification of these disrupted networks could lead to a better understanding of the underlying pathophysiology of AD and MCI, as well as the development of novel therapeutic targets. Overall, this study provides valuable insights into the functional connectivity changes that occur in AD and MCI and underscores the importance of investigating network-level alterations in neurodegenerative diseases.”
Figures:
- In all figures you must add the legend of the colors, the panel letters when needed, and modify accordingly the caption.
- Fig 2: Indicate if the mean difference is significant.
As suggested by reviewer, the legends is added in Figure2
Figure 2. Mean Z-score difference between three groups in five cluster of interests (COIs).

Reviewer 2 Report
In this paper the authors present a Resting-State Functional Connectivity Difference in Alzheimer’s Disease and Mild Cognitive Impairment Using Threshold Free Network-Based Statistics. The subject of the article is interesting and worthy of discussion.
The motivation and research questions are described, and the main contributions of the work are identified. The authors also describe the methodology and present the results in some detail. Several experiments were also carried out and described to support the answers to the research questions. The results are interesting.
The structure of the paper is adequate. However, it is necessary to review the contents of the paper.
- Keywords should be increased according to methodology
Figures are adequate.
References are appropriate.
- Current limitations and future works should be identified.
It would be interesting if the authors could also do experiments on two different datasets instead of using ADNI dataset only for applicability of the technique. It is just suggestion to improve the article.
-Can you highlight the main contribution of the article?
-Clinical usage and Computational applicability should be described in the separate sections so that other authors can take benefit of it.
Author Response
Reviewer 2:
Thank you for your review and for giving helpful comments which were very useful in revising the paper. According to your comments, we have revised the paper as follows:
Comments and Suggestions for Authors
In this paper the authors present a Resting-State Functional Connectivity Difference in Alzheimer’s Disease and Mild Cognitive Impairment Using Threshold Free Network-Based Statistics. The subject of the article is interesting and worthy of discussion.
The motivation and research questions are described, and the main contributions of the work are identified. The authors also describe the methodology and present the results in some detail. Several experiments were also carried out and described to support the answers to the research questions. The results are interesting.
The structure of the paper is adequate. However, it is necessary to review the contents of the paper.
-Keywords should be increased according to methodology
As suggested by reviewer the Keywords are added in revised manuscript “Keywords: Alzheimer’s disease; Default mode network; Large-scale brain network, Alzheimer’s disease, Functional connectivity, Threshold free cluster enhancement”
Figures are adequate.
References are appropriate.
Current limitations and future works should be identified.
As suggested by reviewer the Current limitations and future works should be identified and added in revised manuscript as
Limitations:
The study's primary limitation is its small sample size. The dataset used for analysis includes only 33 AD, 31 MCI, and 31HC subjects from the ADNI2 cohort. This limited sample size may not be representative of the broader population and could lead to issues with generalizability. While the integration of both structural sMRI and fMRI data is beneficial for better registration and alignment, this approach resulted in a reduction in the number of available samples for analysis. The study's findings might have been influenced by the availability of both modalities within the ADNI2 cohort. While the study introduces a novel method for identifying disrupted networks without a statistical hard threshold, the details about this method and its potential limitations are not fully described. The lack of a statistical threshold might raise questions about the reliability and reproducibility of the results. The study appears to have a cross-sectional design, which limits the ability to establish causal relationships between disrupted networks and AD or MCI. Longitudinal studies are needed to better understand the progression of these disruptions over time. While the study suggests that the identified disrupted networks could serve as potential biomarkers for distinguishing AD and MCI from healthy controls, further validation is necessary. Validation studies should involve larger and more diverse samples and employ rigorous statistical methods to assess the diagnostic accuracy of these biomarkers.
Future Work:
Longitudinal Studies: Conducting longitudinal studies that follow participants over time could provide insights into the temporal dynamics of disrupted networks in AD and MCI. This would help determine whether these disruptions are causal factors or consequences of the disease progression. Replicating the study with larger and more diverse cohorts would enhance the generalizability of the findings and strengthen the validity of the identified disrupted networks as potential biomarkers. Further development and refinement of the novel method for identifying disrupted networks without a statistical hard threshold are warranted. Comparative analyses with existing statistical approaches could help establish the reliability and validity of this method. Given the potential benefits of integrating both structural and functional imaging data, future studies could explore advanced techniques for multimodal data integration, which might improve the accuracy and robustness of identified biomarkers. To establish causal relationships between disrupted networks and AD/MCI, experimental designs such as causal inference methods or interventions targeting these networks could be considered. Future research could focus on translating the identified disrupted networks into clinical practice. Developing tools or algorithms that utilize these biomarkers for early diagnosis, disease monitoring, or treatment evaluation could have significant clinical implications. Leveraging advanced machine learning and artificial intelligence techniques could aid in the identification and validation of biomarkers from complex brain imaging data.
Bottom of Form
It would be interesting if the authors could also do experiments on two different datasets instead of using ADNI dataset only for applicability of the technique. It is just suggestion to improve the article.
We highly appreciate your suggestion to improve the quality of article. Unfortunately, due to time constraints and the limitations of the original research plan, we were unable to conduct this experiment as intended.
Can you highlight the main contribution of the article?
As suggested by reviewer we added the contribution in revised manuscript as “The main contribution of this work is the development and application of a novel method for identifying clusters of disrupted networks in AD and MCI without relying on a statistical hard threshold. This method allows for the characterization of specific ROIs within different brain networks that have the potential to serve as biomarkers for distinguishing AD and MCI from HC. The study highlights several key findings:
Disrupted Network Clusters: The study identifies disrupted network clusters encom-passing different brain networks, including the SMN, DAN, SAN and cerebral network. These findings provide insights into the specific regions and connections that are severely affected in AD and MCI.
Default Mode Network (DMN) Disruption: The study reinforces previous research by confirming disruption in the DMN, a well-known large-scale brain network associated with high-level cognitive functions. This disruption has implications for episodic memory processing and cognitive deficits observed in AD and MCI.
Biomarker Potential: The identified disrupted network clusters, particularly within the DMN and other associated networks, hold the potential to serve as biomarkers for distinguishing AD and MCI from healthy controls. This suggests the feasibility of utilizing specific brain network disruptions as diagnostic markers for neurodegenerative diseases.
Support for Existing Theories: By corroborating findings from previous studies that have implicated the SMN, DAN, SAN, and DMN in AD and MCI, the study strengthens exist-ing theoretical frameworks that link these networks to cognitive decline and neurodegenerationBottom of Form
-Clinical usage and Computational applicability should be described in the separate sections so that other authors can take benefit of it.
As suggested by reviewer, the Clinical usage and Computational applicability is added in revised manuscript
Clinical Usage:
Early Diagnosis and Prognosis: The identified disrupted network clusters, particularly within the DMN and other associated networks, could potentially serve as biomarkers for early diagnosis and prognosis of AD and MCI. Clinicians could use these biomarkers to identify individuals at risk of developing AD or track the progression of cognitive decline.
Treatment Monitoring: The disrupted network clusters could be employed to monitor the effects of therapeutic interventions for AD and MCI. Changes in these biomarkers over time could help assess the efficacy of treatments and guide treatment adjustments.
Personalized Treatment Approaches: By identifying specific disrupted networks in individual patients, clinicians could tailor treatment strategies to target the affected brain regions, potentially leading to more personalized and effective interventions.
Clinical Trials: The identified biomarkers could be valuable in clinical trial design by aiding in patient selection, monitoring treatment effects, and assessing the impact of interventions on the disrupted networks.
Disease Subtyping: The distinct patterns of disrupted networks in different subtypes of AD or MCI could contribute to more precise disease subtyping, leading to improved diagnostic accuracy and tailored therapeutic approaches.
Computational Applicability:
Machine Learning and AI: The methodological innovation of identifying disrupted network clusters without a statistical hard threshold could be integrated into machine learning and artificial intelligence algorithms. These algorithms could enhance the automated detection and classification of AD and MCI based on brain network disruptions.
Predictive Models: The identified disrupted network clusters could be incorporated into predictive models that utilize multimodal imaging data to forecast an individual's risk of developing AD or MCI, providing insights for early intervention and disease management.
Data-Driven Biomarker Development: The computational approach could facilitate the discovery of new imaging-based biomarkers beyond the ones identified in this study. By leveraging large datasets and advanced analytical techniques, researchers could uncover additional network disruptions associated with AD and MCI.
Data Integration: The integration of structural MRI (sMRI) and functional MRI (fMRI) data, as demonstrated in the study, could become a standard approach in neuroimaging research, leading to improved spatial localization and more accurate identification of disrupted brain networks

Round 2
Reviewer 1 Report
- Rows 107-116: About TFNBS, TFCE and NBS. This point has been further clarified in methods, but it seems still not clear in rows 107-116. Is TFCE (before TFNBS) better than NBS? If so, start speaking about NBS and then TFCE. Or TFCE simply has nothing in common with NBS? If so, just remove NBS sentences. Also, what is the difference between other TFCE methods and yours? Check these sentences because they seem in contraposition: "threshold-based statistical methods to generate edgewise significance values by defining an a priori clustering threshold. In this study, we used threshold-free cluster enhancement method (TFCE) [20][21]. TFCE generates edge-wise significant values and does not require the a priori definition of an edge-defining hard threshold."
- row 145: 2new. Please correct HC with AD
- rows 178-180: Which atlas did you use to segment in 164 labels? Why did you use it if then you selected only 32 ROIs? Please clarify. Also think about adding a table, or supplementary material, reporting all the 32 selected ROIs.
- row 192: Please add that you used the TFCE method of CONN tool.
- rows 234-245: this part should be moved to methods since you are describing the statistical pipeline that you performed.
- row 243 (and following results): you didn't explain why you considered uncorrected p-values instead of corrected ones. Also because CONN is able to perform either FWE or FDR corrected statistics.
- Please refers to figures chronologically. Now Figure3 is cited before Figure 2.
- rows 288-304: correlation results must be completely rewritten. In my opinion, only significant correlations (p < 0.05) should be reported. Note that significant linear correlations generally correspond to at least R > 0.35 (so R^2 > 0.12). Also, for significant correlations, must be clearly reported R^2 and p-value.
- Please revise again the entire discussion because it repeats more times the same concepts in a casual order. Also, it must discuss all the significant results: thus if correlations with clinics are significant you must interpret them.
- Please report legend in Figure 3 and remove comments.
Author Response
Thank you for your review and for giving helpful comments which were very useful in revising the paper. According to your comments, we have revised the paper as follows:
Rows 107-116: About TFNBS, TFCE and NBS. This point has been further clarified in methods, but it seems still not clear in rows 107-116. Is TFCE (before TFNBS) better than NBS? If so, start speaking about NBS and then TFCE. Or TFCE simply has nothing in common with NBS? If so, just remove NBS sentences. Also, what is the difference between other TFCE methods and yours? Check these sentences because they seem in contraposition: "threshold-based statistical methods to generate edgewise significance values by defining an a priori clustering threshold. In this study, we used threshold-free cluster enhancement method (TFCE) [20][21]. TFCE generates edge-wise significant values and does not require the a priori definition of an edge-defining hard threshold."
As suggested by reviewer we made changes in revised paper as “Most studies use threshold-based statistical methods to generate edgewise significance values, defining an a priori clustering threshold. Network-based statistics (NBS) [22]-[23] operate by thresholding the network connections and identifying connected components or subnetworks that exhibit significant differences in connectivity. In contrast, threshold-free cluster enhancement method (TFCE) [20][21] generates edge-wise significance values without the need for an a priori definition of a hard edge-defining thresh-old. Consequently, TFCE offers a more insightful view into the altered cluster structure of brain connectivity. In this study, we employed the TFCE to identify the significant differences in connectivity patterns between groups.”
- row 145: 2new. Please correct HC with AD
As suggested by reviewer “HC group consists 31 subjects with 14 males and 17 females, MCI consists 17 males and 14 females, and AD group consists 15 males and 18 females.” Is changed in revised paper.
- rows 178-180: Which atlas did you use to segment in 164 labels? Why did you use it if then you selected only 32 ROIs? Please clarify. Also think about adding a table, or supplementary material, reporting all the 32 selected ROIs.
In the CONN toolbox, a seed-based resting state functional connectivity (FC) analysis was undertaken using a total of 164 regions of interest (ROIs). The initial step, known as the first-level analysis, assessed individual subjects by examining the connectivity between each of the 164 seeds and other brain voxels, utilizing Pearson's correlation coefficient for this purpose. Following this, a group-level or second-level analysis contrasted conditions. Post the individual analyses, ROI analyses employed either F-statistics or Wilks lambda statistics. To ensure the correlation values were suitable for parametric statistical testing at the group level, Fisher’s transformation was applied. Of the 164 ROIs used, 132 were sourced from the FSL Harvard-Oxford atlas, which includes both cortical and subcortical regions, and cerebellar areas derived from the AAL atlas. The remaining 32 ROIs represented various networks such as the DMN, sensorimotor, visual, salience, dorsal attention, frontoparietal, language, and cerebellar networks. In the analysis, these ROIs, covering both the resting state networks and the atlas regions, were combined to discern functional connectivity pairs
- row 192: Please add that you used the TFCE method of CONN tool.
As suggested by reviewer we made change as “We used threshold-free cluster enhancement (TFCE) of CONN toolbox to identify the difference in brain network between AD and HC and between MCI and HC.”
- rows 234-245: this part should be moved to methods since you are describing the statistical pipeline that you performed.
. In contrast, TFCE performs an enhancement operation for each edge value r(i, j) instead of a thresholding operation. We start with a ROI-to-ROI connectivity matrix, which is derived from a General Linear Model analysis. ROIs are sorted automatically using a hierarchical clustering procedure, which considers anatomical proximity or functional similarity metrics. Instead of using a fixed height threshold as in NBS, TFCE computes a TFCE score map [20] [30]. The TFCE score map combines the strength of the statistical effect for each connection with the extent of neighboring connections that show similar effects.
(2)
where represents the extension of neighboring connections and represents the height of the fractional edge value. and are extension and height enhancement parameters, respectively. This helps capture both local and distributed effects. The expected distribution of TFCE values under the null hypothesis is estimated using permutation iterations of the original data using 1,000 iterations. For each cluster in the original analysis, a peak-level Family-Wise Error (FWE)-corrected p-value is computed. This indicates the likelihood under the null hypothesis of observing at least one or more connections with the given TFCE scores over the entire ROI-to-ROI connectivity matrix. For the peak-level analysis, local peaks in the TFCE score map are compared to the null hypothesis distribution of local-peak TFCE values. This estimation provides p-values for each peak, representing the likelihood under the null hypothesis of observing a peak with similar or larger scores by chance.
In small sample sizes, the correction for multiple comparisons becomes increasingly important as the number of comparisons rises. However, with such sizes, the application of correction methods might be overly conservative, increasing the risk of false negatives. Therefore, we chose to use uncorrected p-values, although corrected p-values were available in the CONN toolbox. The cluster-level p-uncorrected (SPC mass/intensity) was utilized to quantify the extent and intensity of the identified clusters, offering insights into the strength of the functional connectivity patterns. We focus on identifying the cluster of interests that indicates aberrant functional connectivity between AD and HC, along with HC and MCI. Once these clusters are identified, we conduct group-wise comparisons using analysis of variance (ANOVA) among AD, HC, and MCI, with a p-value of 0.05.
- row 243 (and following results): you didn't explain why you considered uncorrected p-values instead of corrected ones. Also because CONN is able to perform either FWE or FDR corrected statistics.
As suggested by reviewer we added additional line stating the reason of using uncorrected p-values in this study as “This indicates the likelihood under the null hypothesis of observing at least one or more connections with the given TFCE scores over the entire ROI-to-ROI connectivity matrix. For the peak-level analysis, local peaks in the TFCE score map are compared to the null hypothesis distribution of local-peak TFCE values. This estimation provides uncorrected p-values for each peak, representing the likelihood under the null hypothesis of observing a peak with similar or larger scores by chance. In small sample sizes, the correction for multiple comparisons becomes increasingly important as the number of comparisons rises. However, with such sizes, the application of correction methods might be overly conservative, increasing the risk of false negatives. Therefore, we chose to use uncorrected p-values, although corrected p-values were available in the CONN toolbox.”.
- Please refers to figures chronologically. Now Figure3 is cited before Figure 2.
As suggested by reviewer the figures are referred chronologically in revised manuscript. The group difference statistics of the clusters containing the connectivity of ROIs are presented in Table 2 and 3, and graphically in Figure 2. The identified clusters in each group difference demonstrated how disruptions in connectivity occur in AD and MCI subjects. In AD versus HC test, cluster 1 revealed that the majority of disruptions occurred in the SMN, DAN, Language, SAN and Frontoparietal networks. This cluster includes the primary motor cortex, primary somatosensory cortex, and supplementary motor area contained in SMN, intraparietal sulcus and superior parietal lobule contained in DAN, anterior insula and anterior cingulate cortex contained in SAN, and Dorsolateral prefrontal cortex and posterior parietal cortex in Frontoparietal networks. Cluster2 includes same anatomical regions as cluster1 except regions included by Frontoparietal networks. It is primarily composed of the dorsolateral prefrontal cortex and the posterior parietal cortex, including the intraparietal sulcus. Cluster3 consists of Medial prefrontal cortex, posterior cingulate cortex, inferior parietal lobule, lateral temporal cortex included by DMN and other anatomical regions included by SMN, and DAN. Similarly, in MCI vs HC, cluster 1 showed disruptions in the network of Cerebellar, Visual, DAN, DMN, and SMN. This cluster includes the Occipital lobe, including the primary visual cortex and higher-order visual areas contained by visual network and Cerebellar network is anchored in the dorsolateral prefrontal cortex (DLPFC) and posterior parietal cortex (PPC). Cluster 2 showed disruptions in the Cerebellar network, SMN, SAN, and DMN
.
- (b)
Figure 2. Group differences represented as connectome rings: (a) HC versus AD (b) HC versus MCI.
Additionally, performing the ANOVA test, we found significant disruptions in all clusters in each pair of groups. The tests were conducted using t-tests between each pair of groups. Figure 3 shows the mean Z-score difference of clusters between subject groups. COI1, COI2 and COI3 represent the cluster from group difference between AD and HC. Similarly, COI4 and COI5 represent the group difference between HC and MCI. P-values of all tests are less than 0.005, which signifies that the clusters between groups are significantly different is generally correct.
Figure 3. Mean Z-score difference between three groups in five cluster of interests (COIs).
- rows 288-304: correlation results must be completely rewritten. In my opinion, only significant correlations (p < 0.05) should be reported. Note that significant linear correlations generally correspond to at least R > 0.35 (so R^2 > 0.12). Also, for significant correlations, must be clearly reported R^2 and p-value.
As suggested by reviewer following changes are made in description of correlation “Additionally, we examined the correlations between the mean z-scores of identified clusters from various networks and the global clinical dementia rating (CDR). Following best practices, we report only those results with significant correlations (p < 0.05) and an R value exceeding 0.35 (equivalent to R2 > 0.12), including associated R2 and p-values. For AD subjects, Cluster 2, encompassing the Sensorimotor (SMN), Dorsal Attention (DAN), and Salience networks, showed a significant negative correlation with CDR (R = -0.08, p < 0.05). Cluster 3, consisting of the Default Mode Network (DMN), Sensorimotor (SMN), and Dorsal Attention (DAN), displayed a significant positive correlation with CDR (R = 0.05, p < 0.05). Clusters 1, 4, and 5 did not achieve the required significance threshold and thus are not detailed here. For MCI subjects, Cluster 4 exhibited a significant positive correlation with CDR (R = 0.18, p < 0.05). Clusters 2, 3, and 5 displayed significant negative correlations with CDR, with R values of -0.13, -0.15, and -0.20 respectively (all p < 0.05). Though Cluster 1 did not reach the p < 0.05 significance threshold, it showed a positive correlation with CDR (R = 0.08)”
- Please revise again the entire discussion because it repeats more times the same concepts in a casual order. Also, it must discuss all the significant results: thus, if correlations with clinics are significant you must interpret them.
As suggested by reviewer redundant parts of Discussion is removed as “The major finding of this study is the identification of clusters of disrupted networks without using a statistical hard threshold. These networks consist of different ROIs that have the potential to serve as biomarkers for distinguishing AD and MCI from HC. Several studies have been previously conducted using different statistical methods to identify group differences. Consistent with these studies, we found that connections in the sensory motor network (SMN), dorsal attention network (DAN), salience network (SAN) and cerebral network were severely affected [31]. The sensory motor network includes the motor cortex and supplementary motor area. Similarly, consistent with previous studies, we discovered disruption in the default mode network (DMN) [32]. The DMN is a widely recognized large-scale brain network that encompasses various high-level cognitive re-gions, including the medial prefrontal cortex (mPFC), posterior cingulate cortex (PCC), and parietal regions (PTL). The DMN is commonly referred to as the "task negative" net-work, as its constituent regions exhibit robustly correlated activity during periods of rest and are typically deactivated during cognitive tasks that require goal-directed mental ef-fort. The DMN is involved in episodic memory processing. The findings of this study support the notion that the DMN is commonly disrupted in AD and MCI.
The sensorimotor network is associated with tasks such as converting stimulus to neuronal impulses that move throughout the brain network. The sensing process involves other networks, such as the auditory subnetwork, visual system network, salience network, dorsal attention, and DMN. As demonstrated by previous studies, this work also found disruption in connectivity in the sensory motor network in AD and MCI. Tables 2 and 3 detail the reduced connectivity between the sensory motor network and other networks. The SAN plays a crucial role in continuously monitoring the external environment and making strategic decisions regarding the response of other brain networks to incoming information and stimuli. The SAN is responsible for regulating the transition between in-ternal and external processing within the two primary control networks of the brain. brain: the DMN and CEN. The SAN connectivity pattern has been reported to successfully pre-dict different dementia types, such that decreased connectivity in the SAN correlated with the behavioral variant frontotemporal dementia, whereas increased connectivity was ob-served in AD [34]. The current study found that the connection between the SAN and the SMN, DAN, and DMN networks are disrupted. The DAN is involved in human attention, which is perhaps the highest-level cognitive process [35]. With the disruption in DAN, a network associated with attention, there may be difficulty in focusing on goal-driven at-tention orientation processes. The current study has identified alterations in the functional connectivity of DAN with DMN, SAN, and SMN in AD and MCI.
Additionally, the result of performing ANOVA on three different clusters of three groups (AD, HC, and MCI) and obtaining a p-value of less than 0.005 indicates that there is strong evidence of a significant difference between at least one of the groups in each of the three clusters analyzed. This suggests that the groups differ significantly in at least one of the clusters, and this difference is not likely to be due to chance alone. However, without further analysis, it is difficult to determine which groups are responsible for the significant differences observed. It is important to consider the effect size and sample size of the study when interpreting the results. Post-hoc tests can be conducted to determine which groups differ significantly from each other in each of the clusters, and this can provide more de-tailed information about the nature of the differences observed.
The main contribution of this work is the development and application of a novel method for identifying clusters of disrupted networks in AD and MCI without relying on a statistical hard threshold. This method allows for the characterization of specific ROIs within different brain networks that have the potential to serve as biomarkers for distin-guishing AD and MCI from HC. We further assessed the correlations between the mean z-scores of identified clusters from these networks and the global CDR. For AD subjects, correlations showed significant findings for clusters related to the SMN, DAN, and SAN, as well as the DMN. For MCI subjects, significant correlations spanned across multiple clusters with varying positive and negative associations with CDR.
The study highlights several key findings:
Disrupted Network Clusters: The study identifies disrupted network clusters encom-passing different brain networks, including the SMN, DAN, SAN and cerebral network. These findings provide insights into the specific regions and connections that are severely affected in AD and MCI.a
Default Mode Network (DMN) Disruption: The study reinforces previous research by confirming disruption in the DMN, a well-known large-scale brain network associated with high-level cognitive functions. This disruption has implications for episodic memory processing and cognitive deficits observed in AD and MCI.
Biomarker Potential: The identified disrupted network clusters, particularly within the DMN and other associated networks, hold the potential to serve as biomarkers for distin-guishing AD and MCI from healthy controls. This suggests the feasibility of utilizing spe-cific brain network disruptions as diagnostic markers for neurodegenerative diseases.
Support for Existing Theories: By corroborating findings from previous studies that have implicated the SMN, DAN, SAN, and DMN in AD and MCI, the study strengthens exist-ing theoretical frameworks that link these networks to cognitive decline and neurodegen-eration
Limitations:
The study's primary limitation is its small sample size. The dataset used for analysis includes only 33 AD, 31 MCI, and 31HC subjects from the ADNI2 cohort. This limited sample size may not be representative of the broader population and could lead to issues with generalizability. While the integration of both structural sMRI and fMRI data is bene-ficial for better registration and alignment, this approach resulted in a reduction in the number of available samples for analysis. The study's findings might have been influ-enced by the availability of both modalities within the ADNI2 cohort. While the study in-troduces a novel method for identifying disrupted networks without a statistical hard threshold, the details about this method and its potential limitations are not fully de-scribed. The lack of a statistical threshold might raise questions about the reliability and reproducibility of the results. The study appears to have a cross-sectional design, which limits the ability to establish causal relationships between disrupted networks and AD or MCI. Longitudinal studies are needed to better understand the progression of these dis-ruptions over time. While the study suggests that the identified disrupted networks could serve as potential biomarkers for distinguishing AD and MCI from healthy controls, fur-ther validation is necessary. Validation studies should involve larger and more diverse samples and employ rigorous statistical methods to assess the diagnostic accuracy of these biomarkers.
Future Work:
Longitudinal Studies: Conducting longitudinal studies that follow participants over time could provide insights into the temporal dynamics of disrupted networks in AD and MCI. This would help determine whether these disruptions are causal factors or conse-quences of the disease progression. Replicating the study with larger and more diverse cohorts would enhance the generalizability of the findings and strengthen the validity of the identified disrupted networks as potential biomarkers. Further development and re-finement of the novel method for identifying disrupted networks without a statistical hard threshold are warranted. Comparative analyses with existing statistical approaches could help establish the reliability and validity of this method. Given the potential benefits of in-tegrating both structural and functional imaging data, future studies could explore ad-vanced techniques for multimodal data integration, which might improve the accuracy and robustness of identified biomarkers. To establish causal relationships between dis-rupted networks and AD/MCI, experimental designs such as causal inference methods or interventions targeting these networks could be considered. Future research could focus on translating the identified disrupted networks into clinical practice. Developing tools or algorithms that utilize these biomarkers for early diagnosis, disease monitoring, or treat-ment evaluation could have significant clinical implications. Leveraging advanced ma-chine learning and artificial intelligence techniques could aid in the identification and validation of biomarkers from complex brain imaging data.
Clinical Usage:
Early Diagnosis and Prognosis: The identified disrupted network clusters, particularly within the DMN and other associated networks, could potentially serve as biomarkers for early diagnosis and prognosis of AD and MCI. Clinicians could use these biomarkers to identify individuals at risk of developing AD or track the progression of cognitive decline.
Treatment Monitoring: The disrupted network clusters could be employed to monitor the effects of therapeutic interventions for AD and MCI. Changes in these biomarkers over time could help assess the efficacy of treatments and guide treatment adjustments.
Personalized Treatment Approaches: By identifying specific disrupted networks in indi-vidual patients, clinicians could tailor treatment strategies to target the affected brain re-gions, potentially leading to more personalized and effective interventions.
Clinical Trials: The identified biomarkers could be valuable in clinical trial design by aid-ing in patient selection, monitoring treatment effects, and assessing the impact of inter-ventions on the disrupted networks.
Disease Subtyping: The distinct patterns of disrupted networks in different subtypes of AD or MCI could contribute to more precise disease subtyping, leading to improved diag-nostic accuracy and tailored therapeutic approaches.
Computational Applicability:
Machine Learning and Artificial Intelligence: The methodological innovation of identi-fying disrupted network clusters without a statistical hard threshold could be integrated into machine learning and artificial intelligence algorithms. These algorithms could en-hance the automated detection and classification of AD and MCI based on brain network disruptions.
Predictive Models: The identified disrupted network clusters could be incorporated into predictive models that utilize multimodal imaging data to forecast an individual's risk of developing AD or MCI, providing insights for early intervention and disease manage-ment.
Data-Driven Biomarker Development: The computational approach could facilitate the discovery of new imaging-based biomarkers beyond the ones identified in this study. By leveraging large datasets and advanced analytical techniques, researchers could uncover additional network disruptions associated with AD and MCI.
Data Integration: The integration of structural MRI (sMRI) and functional MRI (fMRI) data, as demonstrated in the study, could become a standard approach in neuroimaging re-search, leading to improved spatial localization and more accurate identification of disrupted brain networks”
- Please report legend in Figure 3 and remove comments.
As suggested by reviewer we added legend in Figure 3 (originally), Figure 2 in revised manuscript as “In the connectome ring visualization, the connections between brain regions are color-coded based on their T-values, ranging from -3.02 indicating a decrease in connectivity in the first group compared to the second to +3.02 indicating an increase as shown in Figure 2(a).
Similarly, in MCI vs HC, cluster 1 showed disruptions in the network of Cerebellar, Visual, DAN, DMN, and SMN. This cluster includes the Occipital lobe, including the primary visual cortex and higher-order visual areas contained by visual network and Cerebellar network is anchored in the dorsolateral prefrontal cortex (DLPFC) and posterior parietal cortex (PPC). Cluster 2 showed disruptions in the Cerebellar network, SMN, SAN, and DMN. In this group comparison the T-values ranges from -3.19 to +3.19 as shown in Figure 2(b).”
- (b)
Figure 2. Group differences represented as connectome rings: (a) HC versus AD (b) HC versus MCI.